# Management of the Energy Mix and Emissivity of Individual Economies in the European Union as a Challenge of the Modern World Climate †

**Ireneusz Miciuła** [1,*] , **Henryk Wojtaszek** [2] , **Marek Bazan** [3] , **Tomasz Janiczek** [4] ,
**Bogdan Włodarczyk** [5] , **Judyta Kabus** [6] and **Radomir Kana** [7]

1   Department of Sustainable Finance and Capital Markets, Faculty of Economics, Finance and Management, University of Szczecin, 70-453 Szczecin, Poland
2   Management Institute Management and Command Department, War Studies University, 00-910 Warsaw, Poland; h.wojtaszek@akademia.mil.pl
3   Department of Computer Engineering, Faculty of Electronics, Wroclaw University of Science and Technology, 50-372 Wrocław, Poland; marek.bazan@pwr.edu.pl
4   Department of Control Systems and Mechatronics, Faculty of Electronics, Wroclaw University of Science and Technology, 50-372 Wroclaw, Poland; tomasz.janiczek@pwr.edu.pl
5   Department of Finance, Faculty of Economic Sciences, University of Warmia and Masuria in Olsztyn, 10-720 Olsztyn, Poland; bogdan.wlodarczyk@uwm.edu.pl
6   Department of Logistics and International Management, Faculty of Management, Częstochowa University of Technology, 42-200 Częstochowa, Poland; judyta.kabus@wz.pcz.pl
7   Department of European Integration, Faculty of Economics, VSB-Technical University of Ostrava, 708 00 Ostrava-Poruba, Czech Republic; radomir.kana@vsb.cz
*   Correspondence: ireneusz.miciula@usz.edu.pl
†   This paper is an extended version of our paper published in International Conference Economic Science for Rural Development, Jelgava, Latvia, 9–10 May 2019; pp. 370–378.

**Abstract:** The aim of the article is to present the most important elements to be implemented in the European Union energy policy in the 2030 perspective in the context of sustainable development of the Member States. The solution to the too high emissivity of individual economies in the European Union is the energy mix, which will establish a compromise in the so-called the triad of EU policy goals. This is undoubtedly a current climate challenge for the modern world, which also has a direct impact on the economic situation of EU countries. The basis of the presented considerations and recommendations is a literature review on the subject and a statistical analysis of empirical data of the largest statistical organizations in the EU and the world. The starting point for the analysis is the assessment of the state of the energy sector in the EU. Therefore, the goals and tasks until 2030 result from many conditions of the energy sector. The article provides recommendations for the EU on future climate and energy policy, analysing the practices of member countries empirical and data compiled by the world's largest organizations and institutions, such as the International Atomic Energy Agency (IAEA), the World Nuclear Association (WNA), Eurostat, and the International Energy Agency (IEA). The strategic goals of the EU climate and energy policy presented in the study show the necessary challenges for the implementation of sustainable development in the analyzed sector, which is the driving force of world economies. The conclusions were presented in accordance with the current economic efficiency of various energy sources and the necessity to seek a compromise among the so-called a triad of goals defined in EU policy.

**Keywords:** EU energy and climate policy; economic efficiency; sustainable development management; renewable energy sources (RES); energy sources

---

## 1. Introduction

Energy is the driving force of economies around the world, andaccess to energy sources is one of the basic factors of economic development. This correlation was already visible during the oil crises of the 1970s. Since then, the problem of energy security has appeared in the public awareness. Nowadays, this problem returns and is one of the main topics of discussion in the European Union countries. In addition, changes in the energy market situation, such as fossil fuel depletion in some countries and geopolitical conditions, make the need for EU action stronger than ever. Energy issues are fundamental to the functioning of modern economies. Otherwise, it will be impossible to achieve EU goals in other strategic areas, including the goals of the Lisbon Strategy for Growth and Jobs and the Millennium Development Goals. Therefore, the new European energy policy must, on the one hand, be ambitious, competitive, and long-term, and, on the other hand, it must also be sensible, well-thought-out, and beneficial to all Member States of the European Union. It is therefore essential that the European Union takes up the great energy challenges we are facing today, i.e., increasing dependence on imports, emphasis on access to energy resources, climate change, and access to sustainable, secure energy for all users. The EU is implementing an ambitious energy policy that covers the full range of energy sources, from fossil fuels to nuclear and renewable energy, to trigger a new industrial revolution that will transform the economy into a low-energy economy, while ensuring greater security, competitiveness, and sustainability in the energy we use. Thus, it is becoming extremely important to strike a balance between competing energy challenges. For example, the conflict between creating a competitive energy market and the costly requirements of reducing greenhouse gas emissions is a result of the current technological and resource capabilities. There is also a need to develop a common position as part of an external policy of solidarity. It will be possible only through the development of a coherent strategic climate and energy policy of the EU Member States.

The aim of this study is to present the contemporary climate and energy policy of the European Union and the initiatives of recent years in this field, and to analyze its impact on the conditions for further economic development of the Member States. "Energy policy has been a key area of activity since the inception of the European project. It must now become a priority again." Europe is increasingly dependent on imported oil and gas, while the demand for these raw materials is constantly growing. Therefore, the problem is the lack of diversification of energy sources, as well as the issue of the security of its supply, directly related to the dimension of the EU's external actions. At the same time, Member States face the need to build competitive internal energy markets and increase energy efficiency. Additionally, the Union responds to global problems resulting from the ongoing climate change. Undoubtedly, all EU countries want to pursue the goal of reducing $CO_2$ emissions. However, they differ on how this should be achieved. First of all, a strategy to limit negative climate change must be effective globally and must not significantly reduce economic development and the well-being of societies. Therefore, as part of the EU climate and energy policy, it is first necessary to prioritize the goals that should be achieved in the first place, because all tasks in the short term until 2030 cannot be achieved, e.g., due to limited resources. The nature of the challenges in the area of energy security creates an unprecedented level for enhanced cooperation and co-shaping the EU climate and energy policy. The EU strategy consisting in the diversification of energy sources will contribute to the development of competition and will make it possible to take into account the requirements of environmental protection, and will also become the cause of balancing the interests of energy companies and energy consumers.

The energy mix is a combination of different types of energy production and consumption [1]. Their diversity, that is different ways of producing and using energy, increases the security of the country in case of failure or exhaustion of any of the sources [2]. The energy market functions on an economic basis. Therefore, the presence of greater competition gives the opportunity to choose the energy source according to one's own criteria, and thus favourable prices for end users. When the use of indigenous energy resources is promoted, a country does not have to rely on energy imported from other countries, which has a number of economic and social advantages. Of course, in some cases it will

have to be phased in over a certain period of time due to the necessary investments. However, along with new technological innovations, there will undoubtedly be more and more possibilities. Energy is supplied in the form of electricity, heat, and fuel to operate machinery and equipment. For electricity, the primary energy that is produced from a specific source is then processed to be delivered to end users. In the case of thermal energy, it is similar, but its production can use more alternative energy sources, and its processing and transmission is less complicated. Additionally, there are more and simpler options for supplying individual farms with thermal energy. A model example of an energy mix for heat are the sources used in Sweden (Figure 1), where the share of renewable energy in the energy production balance is 50%, and 30 years ago the share of fossil fuels was dominant (95%), just like now in most countries of the world and the EU. Of course, this is an example of possible diversification of energy sources, which at the same time gives economic advantages (cheaper energy). In other countries, these will be different sources of energy and on a different scale. However, it shows the possibility of increasing the diversification of energy sources in accordance with the principles of sustainable development. Especially, since also in this case, it was previously based on fossil fuels. In addition, this example also shows the time needed to transform the energy market.

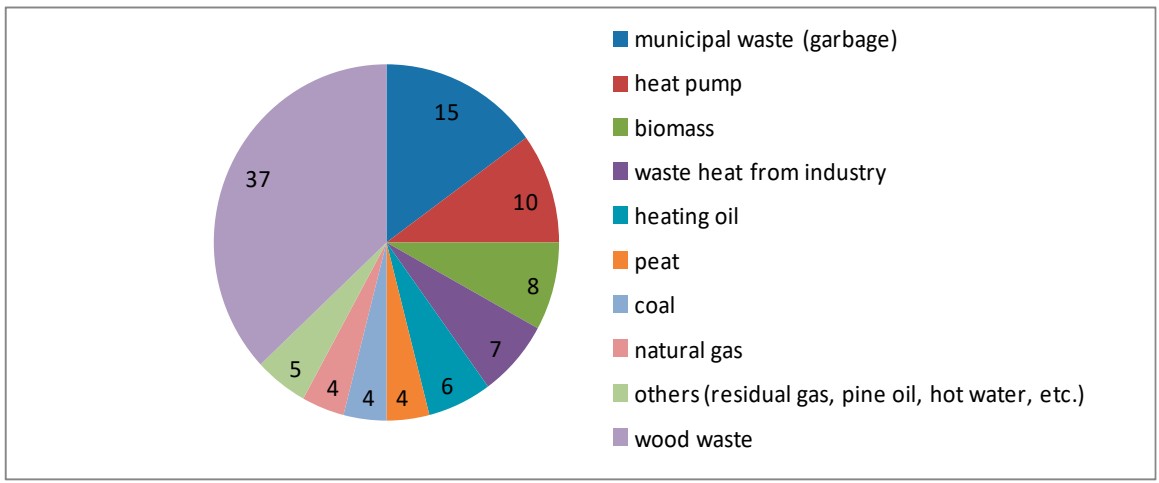

**Figure 1.** Energy sources for the district heating network in Sweden in 2018 (in percent) based on: [3].

Access to energy sources is undoubtedly one of the basic factors of economic development [4]. As part of the energy policy of the European Union, there are many tasks and goals to be achieved. Firstly, European Union countries are increasingly dependent on oil and gas imports, and the demand for these raw materials is still growing. This is a problem due to the lack of diversification of energy sources, which affects the issue of insecurity of energy supply. Secondly, EU countries strive to increase energy efficiency and build competitive internal energy markets. In addition, the Union is responding to global problems arising from climate change. Undoubtedly, all EU countries pursue the goal of reducing $CO_2$ emissions, but they differ in how and when to achieve this. Above all, the strategy of limiting negative climate change must be effective on a global scale. Additionally, it cannot significantly limit economic development and welfare of societies. Therefore, in the framework of the EU climate and energy policy, priority should be given to the goals that are to be achieved first. Because it is impossible to achieve all goals at the same time and in the short term, i.e., until 2030. The reasons are, for example, objective conditions such as limited resources. In this matter, some changes are already visible and certain tasks have been postponed until 2050. The complexity of the issues and the nature of the challenges have been noticed in the fundamental sphere of the economy, which is energy. This creates an unprecedented scope of necessary cooperation and co-shaping the climate and energy policy between EU countries.

The current EU energy policy focuses on the diversification of energy sources, which will contribute, on the one hand, to the development of competition and, on the other hand, to ensuring energy security.

Additionally, it will also constitute the basis for balancing the interests of energy companies and energy consumers. At the same time, this task should be consistent with taking into account environmental protection requirements. Therefore, attention should be paid to the limitations and feasibility of EU plans. In particular, the economic consequences that may occur as a result of the introduction of unfavourable processes forced by the EU. Because the impact of such activities on the functioning of all economic entities (entire economies) will be huge. Therefore, it is necessary to take measures that will allow for the sustainable development of all EU Member States and ensure energy security in a manner based on the principles of rational and efficient use of energy resources. Therefore, it seems that there are no universal rules, but rather separate economies should be considered, for example, due to significant differences in development.

## 2. Literature Review

Energy is the driving force of the world's economies. Access to energy sources is one of the basic factors of economic development [5]. This relationship was revealed during the oil crises in the 1970s. It was then that the problem of energy security arose in the public awareness, which returns today and is one of the main topics of discussion in the EU countries [6,7]. In addition, there have been significant changes in energy markets and geopolitical conditions since then, making the need for action at the EU level stronger than ever [8,9]. Otherwise, it will not be possible to achieve EU goals in other strategic areas, including the goals of the Lisbon strategy for growth and jobs and the Millennium Development Goals [10]. This shows that energy issues are basically fundamental to the functioning of modern economies [11]. Therefore, energy policy must, on the one hand, be ambitious, competitive, and long-term, and, on the other hand, it must also be sensible, well-thought-out, and beneficial to all Member States. It is therefore essential that the EU takes up the great energy challenges we are facing today. The EU is implementing an ambitious energy policy that covers the full range of energy sources, from fossil fuels to nuclear and renewable energy, with the aim of triggering a new industrial revolution that will transform the economy into a low-energy economy, while ensuring greater security, competitiveness, and sustainability of the energy used [12]. Thus, it becomes extremely important to find a balance between competing energy challenges. For example, the conflict between creating a competitive energy market and the costly requirements of reducing greenhouse gas emissions. It will only be possible by developing a coherent strategic policy of the EU countries. The aim of the article is to present the contemporary climate and energy policy of the European Union and initiatives in this field as an analysis of the implementation of challenges for the implementation of sustainable development in the sector that is the driving force of world economies.

The starting point for the analysis is the assessment of the state of the energy sector in the European Union, because the goals of the energy policy in the 2030 perspective result from the current conditions of energy supply and the most important are:

1. Dependence on external energy supplies, i.e., failure to achieve EU energy self-sufficiency. This is due to the excessive energy consumption of economies, limited own resources, and unfavourable conditions of access to external resources. Current estimates show that in 2030 the EU's dependence on external energy supplies will reach 70% [13].

2. Limited freedom to choose the structure of the consumption of energy carriers [14], as the energy sector in the EU is based on coal, and this faces strong opposition from Western European member states. The choice of crude oil and natural gas, in turn, creates dependence on the external market, including from politically unstable countries. Additionally, the priority of renewable resources is a political choice, because in the current conditions, obtaining energy from renewable energy sources is not financially justified.

3. The geopolitical situation, which shows that the current military and political conflicts are taking place in regions rich in energy resources.

4. Limited influence of EU countries on the market of energy resources [15]. The available instruments are foreign policy towards countries supplying energy resources, including strategic investments

in infrastructure and reducing energy demand, mainly as a result of improving energy efficiency by promoting energy saving in buildings and the transport sector, as well as lower losses in energy production and transmission.

5. The propagated plan to reduce the negative impact of the energy sector on the environment and enter the path of sustainable development of the EU Member States and the rest of the world. This is a challenge posed by climate change due to greenhouse gas emissions, a significant cause of which are emissions from energy companies [16].

6. Plan to increase the energy security of EU Member States through the development of competitive fuel and energy markets, including the development of the use of renewable energy and biofuels in transport [17] in accordance with the principles of sustainable development.

The energy policy of EU countries, outlining national strategies in this area, focused on the implementation of three goals [18] (Figure 2):

- minimizing energy prices while ensuring the conditions for self-financing of the sector,
- ensuring an appropriate level of energy security in the short and long term, and
- minimizing the environmental consequences of the operation of energy technologies.

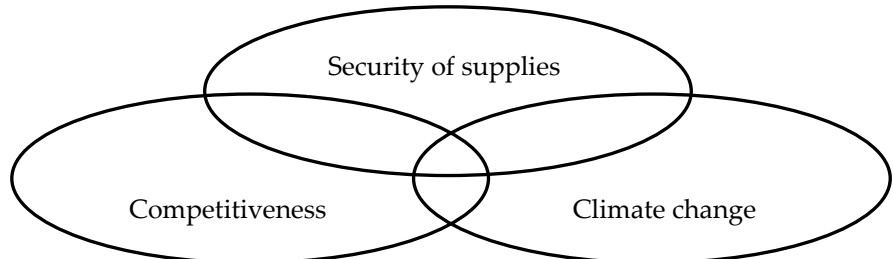

**Figure 2.** The triad of goals of the EU climate and energy policy in the 2030 perspective.

In many fields, however, these goals are mutually exclusive and antagonistic, because both the implementation of greenhouse gas emission reduction technologies and the enhancement of security through the diversification of fuel and energy supplies are costly. With the current state of development, non-emission technologies are particularly expensive, which is not in line with the competitiveness strategy. In addition, taxes, which in the EU, often exceed the price of the energy supplied should be taken into account. An additional problem is the inconsistency of the tools for optimizing energy sources in the short and long term, because although the benefit of introducing market mechanisms to the energy sector is not questioned, they are focused on generating profit and optimizing resource allocation in short time horizons. Therefore, the market is not a source of strategic solutions and the role of the creator of energy policy must be retained by the state or supranational structures. This boundary between the market and the state changes with the evolution of views on the role of strategic sectors, among which the energy sector plays a special role as a fundamental sector, having an impact on prices in the economy, and thus directly on the economic development of the Member States. As a result of the double crisis (financial and political—energy), and due to the controversy and the lack of compromise among the EU Member States as to the effectiveness and importance of the costly effects of climate protection, economic priorities have been recognized as the most important. Therefore, at present, the European Commission has expressed the conviction that economic competition should play the greatest role in the process of energy market integration. However, as part of the integration of the EU energy market, there are many problems to be solved, including fiscal differentiation, lack of technical harmonization of infrastructure, and different approach of EU Member States to environmental protection [19]. The EU is increasingly dependent on imported oil and gas, while the demand for these raw materials continues to grow. Therefore, the greatest challenges for EU Member States include the lack of diversification of energy sources, as well as the issue of security

of supply, which is directly related to the external actions undertaken by the EU in relation to the most important suppliers of energy resources. At the same time, Member States face the need to build competitive internal energy markets and increase energy efficiency. Additionally, the EU responds to global problems resulting from the ongoing climate change. Overcoming these problems has become a priority in the EU's energy and climate policy.

The goals of the EU in the energy policy to be achieved by the 2030 perspective in the context of sustainable development of the Member States are as follows:

- Security of energy supply—The EU is increasingly exposed to the effects of fluctuations and rising prices in international energy markets and to the consequences of the increasing concentration of energy resources among the world's few countries. As part of increasing security of energy supply, the EU is taking steps to reduce its vulnerability to external influences resulting from import dependence. That is why the EU promotes the use of its own, available energy resources and investments in renewable energy sources, and on the international market it undertakes measures to diversify the directions of energy supply. For political and economic reasons, it is unacceptable for some EU Member States to be completely dependent on supplies from one direction only (e.g., from Russia).

- Competitiveness and the EU internal energy market—the aim is to create an internal energy market by implementing directives on the liberalization of the energy sector. As a result, competition will increase, which will result in price reductions and stimulate investment [20]. A single market in energy and the competitiveness of producers and distributors is essential to supporting a common European energy strategy. Therefore, the primary task is to eliminate administrative, technical, and other barriers to trade in energy services in order to enable the development of the EU internal energy market. A great challenge in this matter is the appropriate legislative framework that will create fair conditions for functioning for all EU countries.

- Diversification of energy sources—is related to the concept of an energy mix which is a mixture of various types of energy. Their diversity increases the country's security in the event of a failure or exhaustion of one of the energy sources. An additional aspect of creating the possibility of choosing an energy source is the functioning of an integrated EU market based on economic competition. By promoting one's own energy resources, a positive aspect is the independence from imported energy, which has broad economic and social advantages. In the 2030 perspective, the EU supports the diversification of energy sources, but, first of all, focuses on climate-friendly resources. This increased the importance of renewable energy, the share of which in total energy consumption in 2010 reached 12.7%. The European Commission maintained the binding target that by 2030 the level of renewable energy sources in the overall balance of energy consumption in the EU was 27% [21]. For some of the targets set, the EU is aware that these values will not be achieved, especially with the current loosening of climate policy to support competitiveness and security of energy supply. On the other hand, with regard to coal and nuclear energy, the EU has not taken specific decisions as to the numerical target, and in addition, climate policy instruments (taxes, trading in $CO_2$ emissions) have a negative impact on the competitiveness of obtaining energy from coal on the EU market. On the other hand, the decision to develop nuclear energy was left to the discretion of the Member States.

- Increased energy efficiency—means lower energy consumption while keeping the level of economic activity unchanged [22]. Energy saving is a broader concept than efficiency because it also includes reducing consumption by changing behavior or reducing economic activity. The main goal of improving energy efficiency is the pursuit of zero-energy economic growth, i.e., economic development without an increase in the demand for primary energy. Increasing the efficiency of energy use has a great potential for use in the production and distribution of energy. The EC emphasizes the strong relationship between energy efficiency and environmental protection. Although the achievement of the goal of reducing the energy consumption of the economy by 20% by 2020 has been postponed to 2030, it is one of the few tasks that all EU countries are willing

to implement [23]. Achieving this goal will mean savings of around EUR 100 billion per year and a reduction of $CO_2$ emissions to the atmosphere by 800 million tonnes per year.

- Sustainable development—this goal can be defined as the willingness to seek instruments that will ensure a balance between the objectives of environmental protection, competitiveness, and security of supply. This is manifested by ensuring the continuous, sustainable development of the energy sector by increasing efficiency and safety standards, extending the availability of various energy sources, increasing competitiveness, and reducing greenhouse gas emissions [24].

- Research and development of innovative technologies for energy production and transmission—invest in technological innovations in the energy sector that will lower costs and increase the efficiency of energy production. Renewable energy sources (RES) are the future for further research on a technology that will reduce the costs of its implementation and increase efficiency in energy production [25]. The example of vertical wind farms shows that this is the right development path for energy generation. Another interesting example is the capture of $CO_2$ by microalgae, which not only avoids $CO_2$ emissions but also enables the production of valuable by-products [26,27]. This type of research confirms the possibility of technological innovations that can lead to the invention of new and economically effective energy sources [28]. In principle, the development of innovation applies to all energy sources, including low-emission coal and gas technologies and generation IV nuclear reactors. These investments are also important to ensure that Europe remains a world leader in energy technology. The instruments of implementing this goal by the EU include R&D projects, grants, and competitions for energy innovations.

- Solidarity in external policy—the aim is to establish mechanisms supporting solidarity among EU countries. However, the establishment of specific instruments is still at the stage of consultation between Member States. Additionally, there is no consensus among EU member states on how strong and deep the common external energy policy should be. On the other hand, solidarity in external policy is the foundation for the realization of other EU goals.

- Energy infrastructure—is a kind of "bloodstream", without which it is impossible to achieve other goals. Integrated and reliable energy networks are a prerequisite for achieving the goals of the EU's energy and economic policy. The development of energy infrastructure will ensure a properly functioning internal energy market, guarantee security of supply, enable the integration of RES and increase energy efficiency. Among the priorities to be implemented in the 15-year perspective (until 2030), the European Commission lists:

  - energy corridors important for Central and Eastern Europe,
  - strengthening connections between national systems,
  - connection with wind farms in the North Sea and the Baltic Sea, and
  - strategic infrastructure projects for gas nodes in the Middle East (Nabucco and White Stream project).

The implementation of energy policy objectives requires EU Member States to undertake investment and modernization measures, especially in the field of energy infrastructure. Infrastructure is a critical element for the proper functioning of the entire energy market, and it has the greatest impact on the efficiency and effectiveness of this market. The need to invest in energy infrastructure results both from the insufficient extent of its expansion, technical condition, or the age of individual facilities, as well as from the projected increase in consumer demand for energy and the need to ensure the security of its supply. The implementation of the objectives of the European energy policy described in the article is to lead the EU to achieve an economy with a lower consumption, safer, more competitive, and sustainable energy. On the other hand, as a result of the EU's willingness to achieve so many, often contradictory, goals, the process of crowding out industrial enterprises from the Member States should be noted. This process is due to high energy prices, and therefore also high production costs, in many industrial sectors in EU countries. As a consequence, the industry of EU countries

is losing its competitive position on a global scale, which reduces employment in industry and the business environment. The energy goals to be achieved in the first place are to ensure the security of strategic supplies, the proper functioning of the internal energy market, the reduction of greenhouse gas emissions, and, perhaps above all, to reaffirm one common voice of the EU on the international stage. Achieving these goals is to contribute to transforming Europe into an economy with high energy efficiency and low $CO_2$ emissions. This will usher in a new industrial revolution, accelerating the transition to low-$CO_2$ economic growth and, in the longer term, to a significant increase in production and consumption of low-emission locally produced energy. Energy issues must become a major part of EU relations with countries external to EU Member States, as they are necessary to ensure geopolitical security, economic stability, social development, and the fight against climate change.

## 3. Results

Figure 3 shows the percentage breakdown of the world's most used energy sources. World data shows the dominant position of the three energy sources. Because oil, coal, and gas cover 86.6 percent of all energy needed in the world. In contrast, renewable energy sources cover only 8.3 percent of the world's energy needed. However, the great potential of these energy sources should be noted, because oil owes such a large share to its use mainly in transport, where technological innovations allow their use. However, for electricity coverage, its share is only 5.9 percent, which can be seen in the next chart. In Figure 4, we have data on the use of specific energy sources for electricity production in the world in 2017. For electricity production, we see the dominance of coal as a source of this type of energy. Coal and gas account for 63 percent of the world's energy demand. Both of these energy sources are also dominant in the European Union countries, providing 59.6 percent of energy. Figure 5 contains an analysis of the energy mix for electricity production in the European Union countries. The structure is slightly different due to the depletion of coal resources in the western countries of the EU, which results in coal use 10.2 percent less than in the world. However, it also causes greater use of gas from the outside, which is a problem of dependence on supplies. Therefore, the natural direction is to use other energy sources, especially those available locally. It is a good direction for sustainable development. However, when analysing data, the transformation process is not quick and takes time. It is also due to the fact that new technological possibilities are just emerging, which will not be economically effective immediately. Despite the willingness to meet the demand with domestic resources, which is reflected in a greater share of nuclear energy (by 5.5 percent) and other renewable energy sources (by 2.7 percent), the share of gas needed to ensure energy supplies in EU countries is 7% higher than in world. In the case of oil and hydropower, their share is higher in global electricity production than in the EU, due to countries with oil-rich resources and natural geographic conditions conducive to the use of hydropower.

Figure 6 presents a detailed summary of the energy mix in individual EU countries in 2017. The EU-25 average shows changes in the main energy sources used to produce electricity. Gas comes first, which means that European Union countries are highly dependent on external supplies of this raw material. The second place is taken by coal, which is dominant in the Central and Eastern EU countries. However, due to the exhaustion of this raw material in Western countries, its use has decreased naturally. It can be said that in terms of energy, the countries of Central and Eastern Europe are in the place where Western Europe was several years ago. Among other things, this fact confirms the thesis that the EU energy and climate strategy should be divided into several groups of member states. Due to the developmental differences, one universal approach to energy and climate policy will not function well (without harming other important economic and social areas). By analysing the presented data, it is possible to determine the situation and energy strategy of individual countries. Many EU countries, due to the depletion of coal resources and the intention to use their own resources without dependence on external gas suppliers, choose to develop nuclear energy. These countries include France (75%), Lithuania (76%), Slovakia (54%), Belgium (52%), Hungary (43%), and Sweden (38%). A large growing share of nuclear energy in the energy mix should be noted in such countries as

Bulgaria (36%), the Czech Republic (35%), Slovenia (34%), and Finland (32%). However, also when it comes to this source of energy, there are many concerns, especially with regards to safety in the event of a nuclear power plant accident. This often inhibits the development of nuclear energy. For example, such are the plans, for example, in Germany, where its share is 21%. This is all because of concerns about the nuclear disaster in Fukushima in Japan. It was planned to replace this source of energy with renewable energy sources, but due to insufficient power, the production of energy from coal, including lignite, was also increased. Therefore, now the plans are at a crossroads, because to ensure security of energy supply, it will be necessary to increase energy production from coal or nuclear power. Further, countries such as Italy, Greece, and Portugal do not produce nuclear energy due to the lack of public support. Others, such as Poland, plan to launch nuclear power plants, but it takes years to materialize. Denmark and the United Kingdom have abundant gas and oil resources, but still have a significant share of coal-based energy in the energy mix, 48% and 28% respectively. When it comes to hydropower, it is highly recommended to use. However, it depends on favourable natural conditions. These are mainly found in Iceland and Austria, which allows the production of most of the energy in hydropower plants, 71% and 63% respectively. Other sources of energy, especially renewable ones, should be developed thanks to the emergence of technological innovations that will allow for economically effective use, which will allow for an increase in their share in the energy mix.

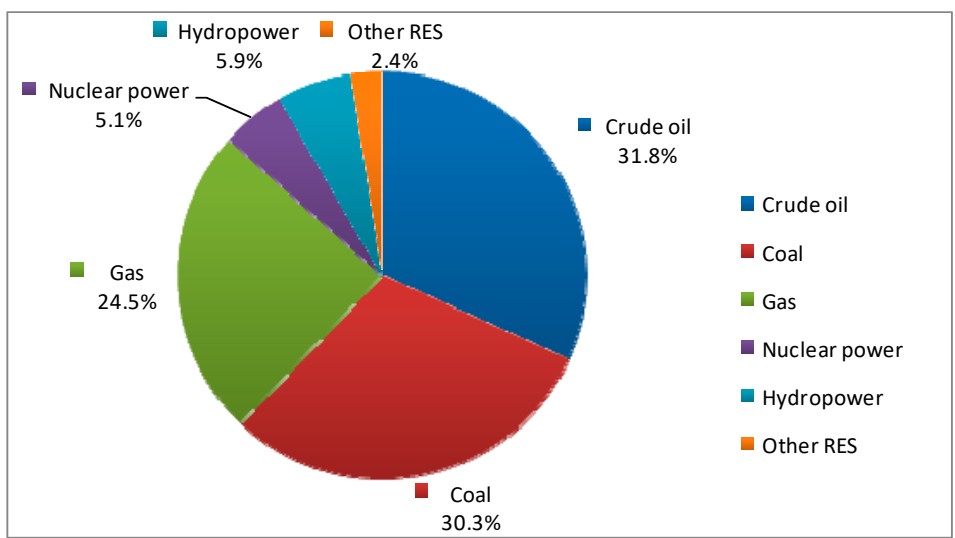

**Figure 3.** Percentage share of the use of specific energy sources in energy production in the world in 2017 [29].

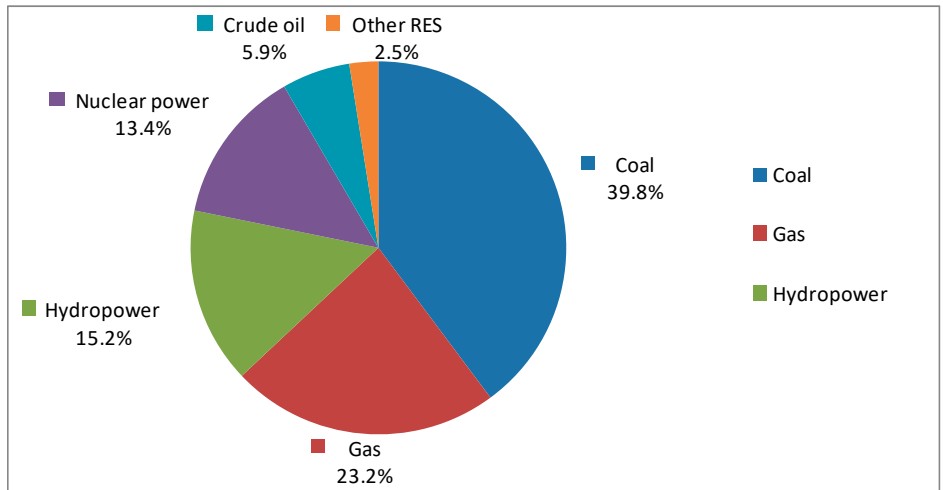

**Figure 4.** Percentage share of energy sources producing only electricity in the world in 2017 [29].

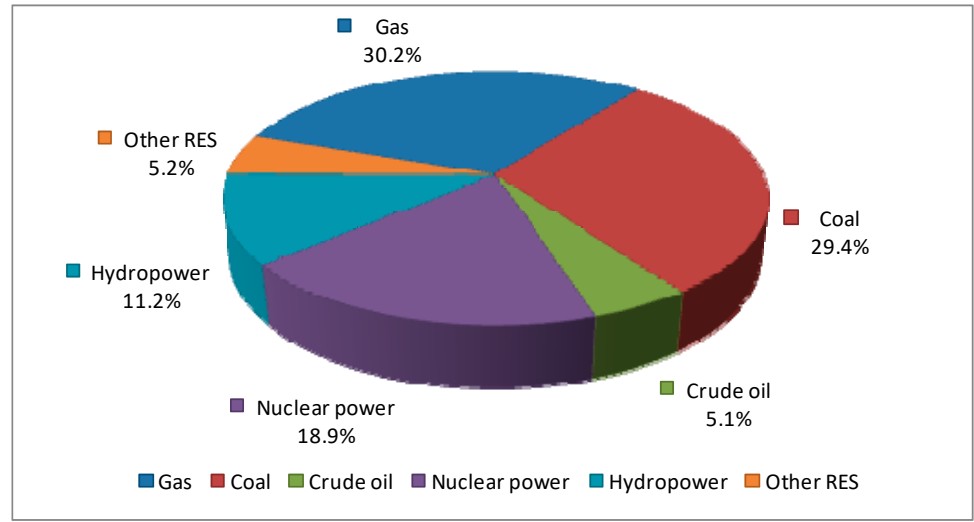

**Figure 5.** Percentage energy mix for the production of electricity in the European Union in 2017 [13].

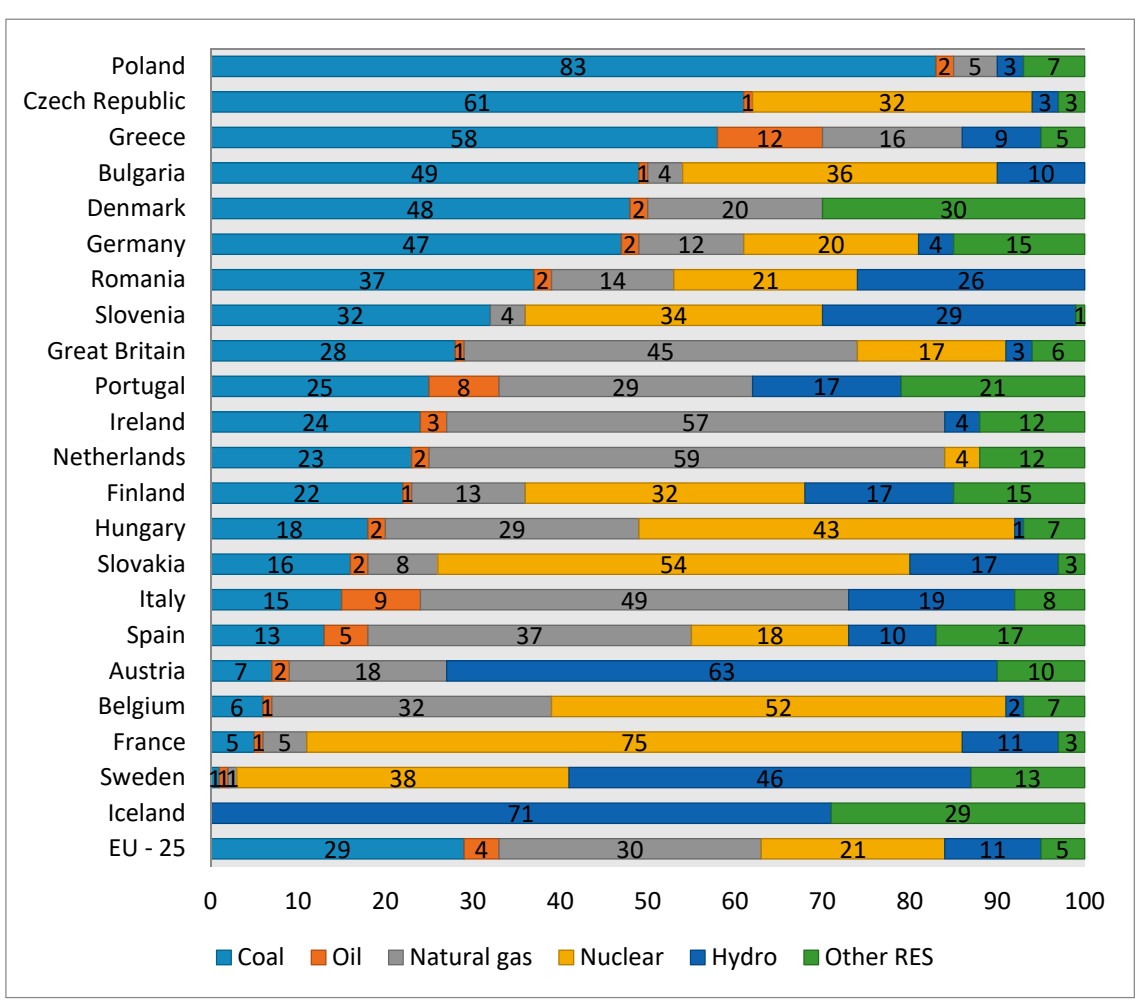

**Figure 6.** Percentage energy mix for electricity production in EU countries in 2017 [30].

Huge investments in renewable energy sources do not translate into a significant increase in the share of these sources in electricity production. Their share in the EU-25 is only 5 percent. As a state, Germany is primarily responsible for the direction of energy and climate policy throughout

the European Union. When analysing the energy situation of EU countries, it is clear that this is a wrong decision, where one country imposes solutions on all other EU members through the prism of its own situation. Especially, where the situation in Germany reveals many conflicts which, as a consequence, may lead to abandoning some goals in relation to others. Despite numerous controversies, an operational program was adopted, the aim of which is to reduce $CO_2$ emissions in line with previous EU plans. However, now Barbara Hendricks (SPD), the Federal Minister for the Environment, Nature Conservation, Construction, and Nuclear Safety, concluded that it would not actually be implemented due to the plan to completely abolish nuclear energy. In this case, due to the lack of alternatives, there will be a renaissance of coal-fired power plants. They show the data of the world leader in lignite mining (Germany), which plans to increase electricity production from this raw material. Therefore, the countries of Central and Eastern Europe, which have greater coal resources than those of Western Europe, also want to increase their coal extraction so that it can be exported to EU countries that are already increasing imports of this raw material from non-EU countries. Countries such as Ireland, The Netherlands, Belgium, Italy, Spain, and Portugal choose gas as their main energy resource due to easy access to external markets, including gas deposits owned by Great Britain and Denmark. However, gas is also high in greenhouse gas emissions, like other major energy sources, its emissions are high in greenhouse gases. Review of greenhouse gas emissions by energy source is shown in Figure 7. The double bars in the figure correspond to the minimum and maximum emissions for the different technologies that can be used within the given energy source. Undoubtedly, the constantly emerging technological solutions have a large impact on the emission of greenhouse gases and economic efficiency. For example, new coal-gas technologies, such as CCGT (Combined Cycle Gas Turbine) and CCS (Carbon Capture and Storage), significantly contribute to the reduction of greenhouse gas emissions, and this is similar to those associated with RES. This shows that the technology used for production and consumption is of great importance when choosing an energy source [31]. A given technology can completely change the optimal choice of an energy source in terms of the most important selection criteria. Because technological innovations affect the availability and efficiency of obtained energy as well as financial costs and side effects (greenhouse gas emissions).

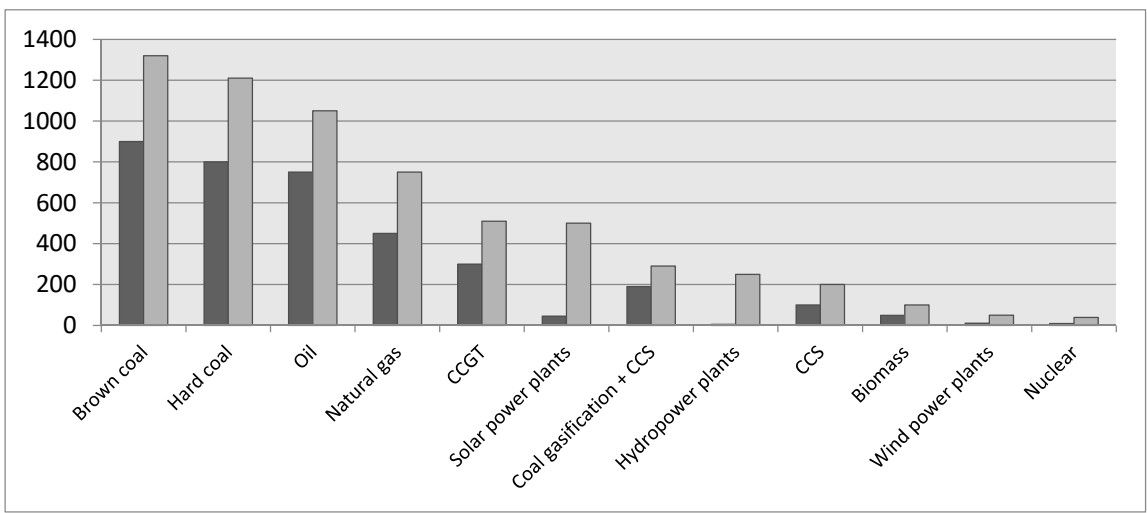

**Figure 7.** Overview of greenhouse gas emissions by energy source (grams per kWh) [30].

　　For transport, crude oil and its derivatives are the dominant energy resource. Due to the high emissivity of oil in transport, there is a great potential to reduce $CO_2$ emissions. Statistics show that transport is responsible up to 20% in the entire basket of sectors responsible for $CO_2$ emissions. The emergence of new technological opportunities will result in greater opportunities to diversify energy sources. For example, it is predicted that in the 21st century a new energy mix will be created for the operation of machines and devices. However, the process appears to be slow due to the bottlenecks

in the development of alternative fuels used by the oil lobbies. Another reason is the fall in oil prices on the market. It is due to growing exports from Russia and the USA, as well as increasing production in Africa and North America. These changes are dynamic and of great importance, which is confirmed by the fact that the share of OPEC countries in the global oil market has decreased from 50% to 30% [32].

Figure 8 shows data on global $CO_2$ emissions in 2006 and forecasts of their increases until 2030. The share of EU emissions in the global $CO_2$ emissions was only 3 billion tons, which is twice as low compared to the USA. This fact clearly shows that the entire EU has no significant influence on global emissions, despite the fact that Germany ranks sixth in terms of global $CO_2$ emissions. Therefore, actions aimed at reducing $CO_2$ emissions in the EU Member States will not affect global warming.

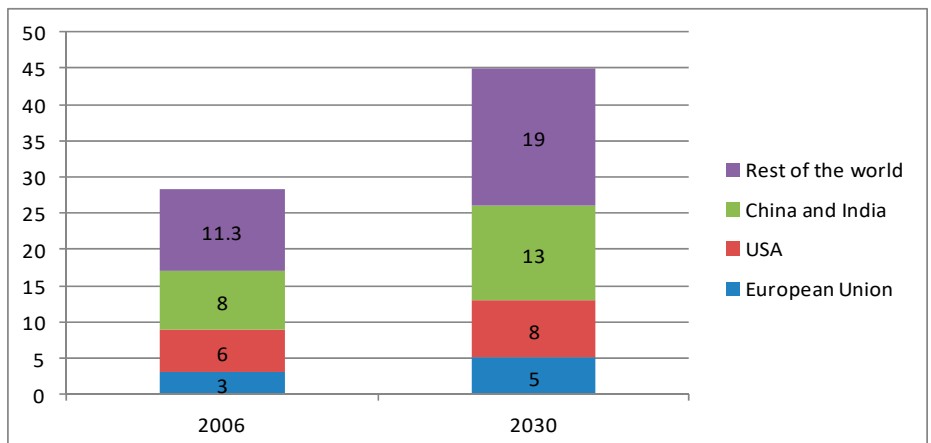

**Figure 8.** $CO_2$ emissions (billion tons) [33].

Figure 9 shows the top ten countries responsible for the highest $CO_2$ emissions. The percentage of carbon dioxide emissions in relation to the whole world (100%) is given in brackets. Analysing the data, it can be seen that the first eight countries are responsible for almost 60% of global carbon dioxide emissions. Germany (2.9%) is the only country in the European Union in the world ten. The next EU country in terms of $CO_2$ emissions is Great Britain (1.4%) and is in the fourteenth place in the world. Then, selected EU countries are shown for comparison. For example, the emissions generated by Poland (0.9%) represent less than 1% of the world market. This shows no impact on reducing global warming in the absence of international agreements.

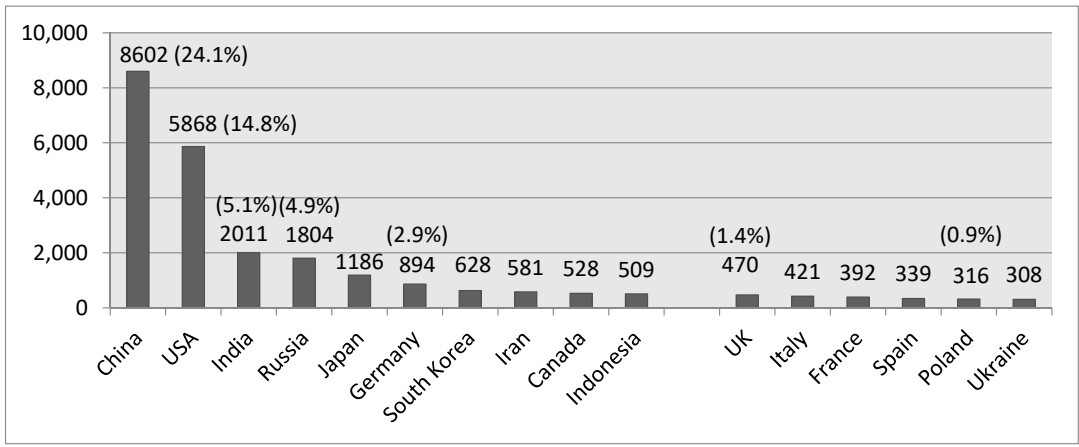

**Figure 9.** $CO_2$ emissions in the top ten countries and selected countries in 2017 (in millions of Mg (Megagram [the symbol Mg] is the standard unit of mass in the International System of Units. It is equal to one million grams [commonly known as a ton])) [30].

Additionally, it seems that in order to reach a compromise with the EU triad of goals, increasing energy efficiency is more important for reducing $CO_2$ emissions. Energy efficiency effects can be verified with indicators that show the relation of emissions to GDP or population of a particular country. Of course, for the right conclusions on the basis of which strategies in energy and climate policy for selected groups of countries will be developed, detailed analyzes of many indicators in this area should be carried out. The Figure 10 presents the $CO_2$ emission rate per capita.

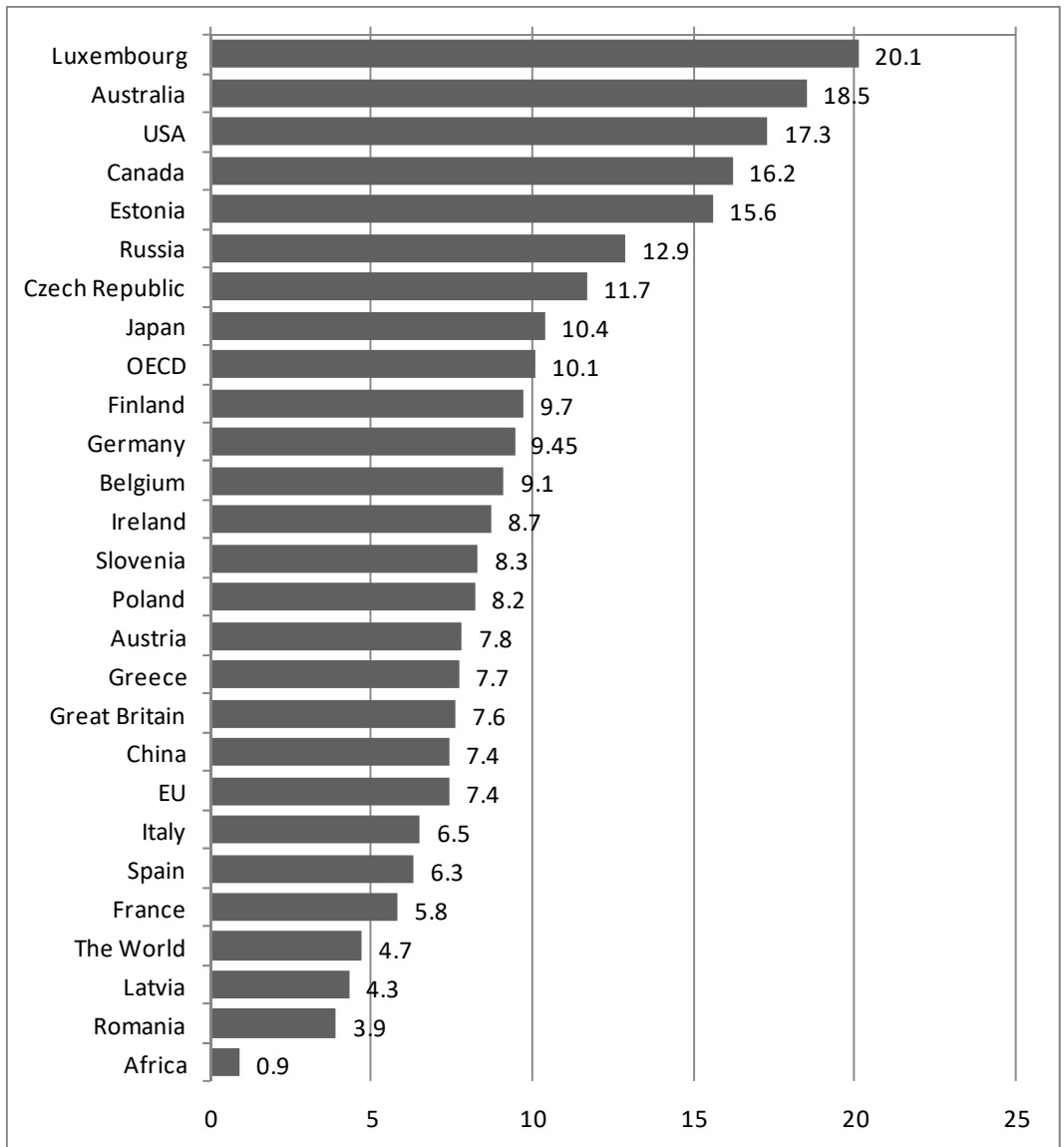

**Figure 10.** $CO_2$ emissions in selected countries in 2017 (tons/capita) [24].

The first place in an economy such as Luxembourg (10 million tons with a population of less than half a million) and Estonia's fifth do not really matter much and are small values that can change in a very short time. However, they are gaining importance for medium and large economies. It is worth noting that China in this case is characterized by a lower rate than the EU average, and India has two times lower than the world average of 4.5 tons per capita [34]. Germany has a high value close to the OECD average of 10 tonnes per capita. All data clearly shows that the German government and economic associations pointing to Poland as the leading producer of "dirty" energy to blame for the failure of the EU's climate policy, including the new rules for trading $CO_2$ emissions, is a mistake, which is even emphasized by institutions and media in Germany. Der Tagesspiegel points out that

the German government, by adopting the EU climate program at the last moment, has taken a step backwards, and has been allowed to avoid international embarrassment; this shows that Germany is not convinced of the EU policy it is pushing. An interesting observation from the analysis of the Institute from Copenhagen is the statement that "$CO_2$ is a by-product of economic growth or $CO_2$ and GDP (Gross Domestic Product) are lovebirds", which is confirmed by the data on the one hand on developed countries and on the other African countries. Another good example confirming this thesis about the modern world and showing the need for multidimensional analysis is the position of Great Britain, which reduced $CO_2$ emissions by 14%. However, this only applies to UK emissions, and it has not been noted that most UK emissions are currently imported, mainly from China. So, if we include the imported emissions in the balance sheet and the exports are deducted, it turns out that the UK has increased $CO_2$ emissions by 18% in the last 20 years. In connection with the above analysis, undoubtedly, the EU energy and climate policy will have to change, and the period between 2020 and 2030, i.e., the practical duration of the operation of most instruments and accounting for the adopted goals, will be decisive whether there will be a complete change in EU policy. In the future, goals for the EU's energy and climate policy should result from the analysis of a series of statistical data and indicators that will rationally take into account all the goals included in the triad of EU goals. This will provide a long-term and predictable framework for investment in the further development of diversified energy sources on a market basis, with sound environmental considerations.

Due to the limited reserves of conventional resources, most of the Member States of the European Union directly face a major problem, namely, the insecurity of energy. Indirectly, also because of this and the inability to provide energy from other sources, the European Union is facing dependence on external suppliers. Especially from one supplier, which is Russia. In this matter, it is certainly necessary to develop an infrastructure that will ensure supplies from other sources, which will bring many positive changes in economic and political matters. Competitiveness, as another goal of the European Union countries, would allow the reduction of energy production costs. Unfortunately, legal norms often created protect the interests of specific countries or industries. Therefore, for the policy to be coherent, the same legal norms should be created for all economic entities associated within the European Union and there is a need to define priorities. Above all, energy security for the economy must be ensured. This means that each country should have the necessary amount of energy at market prices (competitiveness). Unfortunately, these prices often depend on international policy. The solution to this problem would be a full association of European Union members in terms of price negotiations with external energy suppliers. Especially since the energy market is constantly changing. Additionally, the actual prices are often those that are charged at the time of purchasing energy. Therefore, it is necessary to create a market based on economic principles, which will favour the creation of prices under long-term contracts. Therefore, in order for the financial aspects to be of decisive importance, projects are implemented that allow for obtaining new sources of supply and transmission of resources and the diversification of energy sources [35].

Figure 11 shows an analysis of resources that meet 80% of a country's energy demand. It covers solid fuels, oil, and gas. As already noted, the situation of individual EU countries is different despite the growing dependence on external supplies in the entire EU association. For example, Denmark is completely energy independent because of its rich natural resources (the main exporter of oil and gas in the EU). At the same time, it also has the largest share of wind energy production in the EU, thanks to favourable conditions in this area. It only imports solid fuels for financial reasons only. The situation is similar in Great Britain due to the access to oil and gas resources from the bottom of the North Sea. In this case, the dependence on imports is only 10%. On the other hand, a group of countries (Poland, the Czech Republic, Estonia, Romania, Bulgaria, and Slovenia), due to their natural reserves of solid fuels (mainly coal), are more independent than in other Member States. Other examples are countries that, wishing to reduce dependence on external supplies, develop their own energy production, focusing on its specific source. For example, France for nuclear energy, and Sweden for a diversified energy mix of available opportunities, including renewable sources.

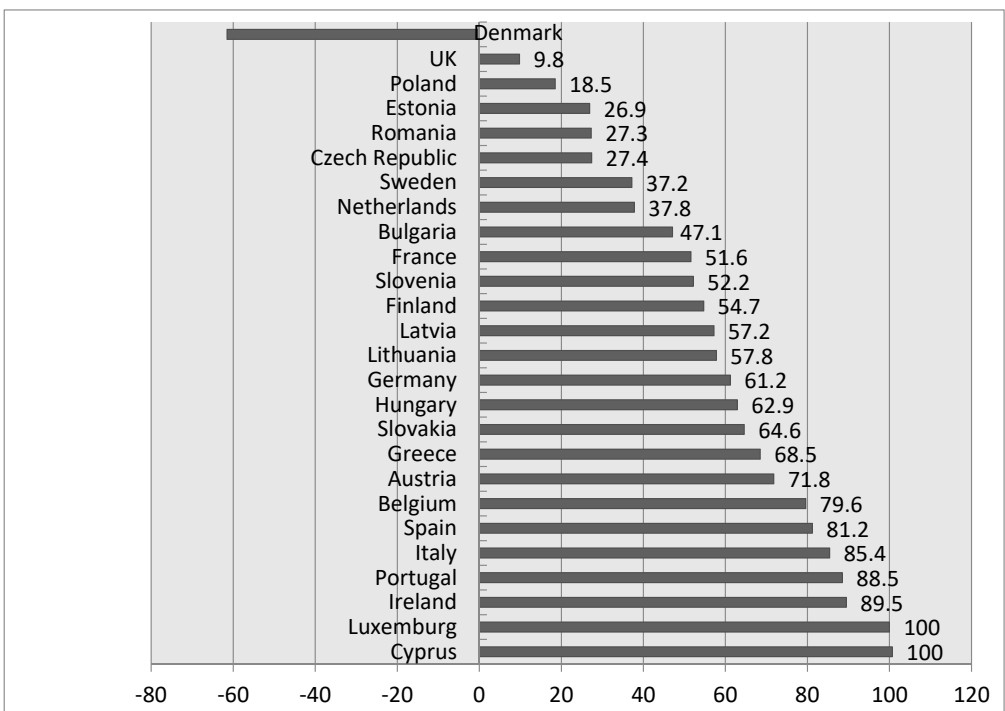

**Figure 11.** Percentage dependence of EU countries on imported conventional fuels in 2018 [30].

Due to the energy needs of the largest EU economy, Germany generally achieves 60% dependency. Poland is distinguished from other countries by large coal resources, including hard coal, which in general gives a low ratio of 18%. However, due to internal problems in the management of the coal industry and the EU climate policy, coal production is decreasing, which may result in the need to increase external supplies. Additionally, other countries with lignite resources, what we see in Figure 12, seem to be at a crossroads as a result of EU policy, which is confirmed by the fact that this raw material still has a large share in energy production, with simultaneous investments and already significant shares of nuclear power plants in the energy mix of these countries.

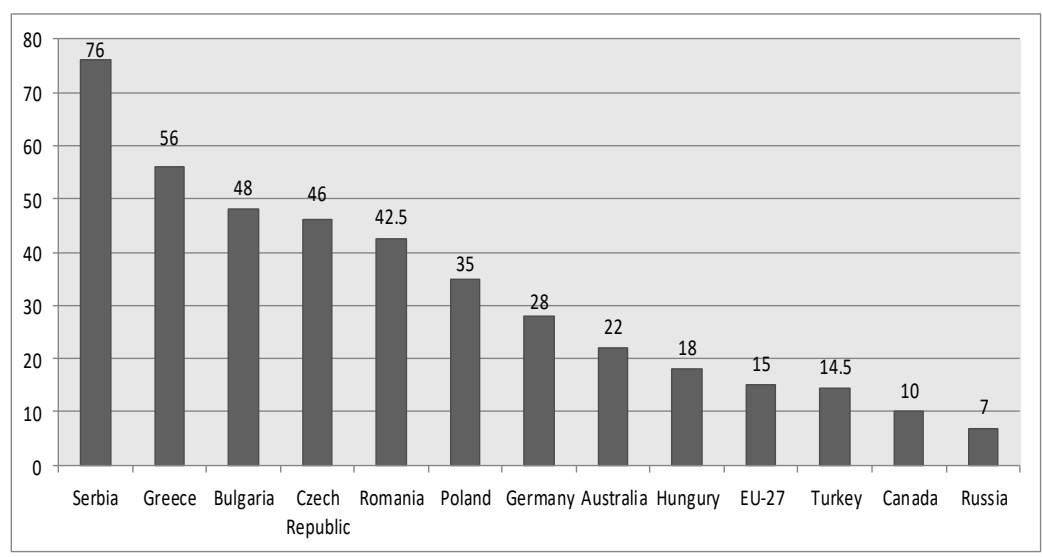

**Figure 12.** Share of lignite in electricity generation in a given country in 2018 (percent) [36].

Therefore, it would be rational from a financial point of view, with simultaneous possibilities of using own energy resources, for the development of obtaining energy from brown coal.

Germany implements this assumption by investing in coal-fired power plants, being at the same time the world leader in lignite mining in the world, in contradiction to the policy it is pushing in the EU arena. Because lignite is the most emitting source of energy. The solution is to use new carbon technologies (CCS), which are able to reduce $CO_2$ emissions to values comparable to RES [37]. Therefore, state associations (clusters) should be established for mutual benefit for the development of a given energy source. The same is true for other resources owned by other countries, e.g., gas by Denmark, Great Britain, and the Netherlands. Cooperation with stable and predictable external partners, such as Norway, would also be beneficial. This is particularly important in the face of increasing dependence on external supplies and the inability to satisfy them with our own resources (Figure 13).

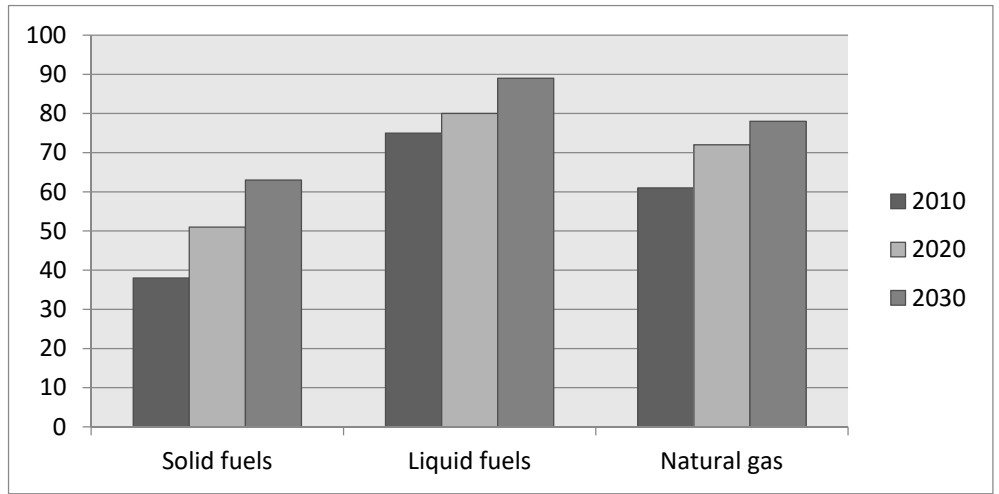

**Figure 13.** Percentage share of imported fuels in the total demand for energy products in the EU-25 and the European Commission's forecast for 2020 and 2030 [30].

Sweden is a good example of how to solve the problem of dependence on external supplies when there was a crisis in the commodity markets causing significant increases [38]. Self-sufficiency, achieved for electricity and heat supplies, has become a priority, as transport should be excluded. 95% of electricity is obtained from hydro (48%), nuclear (38%), and biomass (8%) plants; heat supply is a model for the energy mix of energy sources, including the incineration of waste (garbage).

The strategic aspects of energy security in the new geopolitical situation require the implementation of key investments [39], such as the Nabucco project, for the diversification of supply directions and the possibility of selecting energy sources on market principles. The importance of these investments is demonstrated by the dependence on Russia, which creates a lack of security and competitiveness. Therefore, you need to use your own resources and access the entire market. The current state of technology does not provide an economic justification for the global promotion of renewable energy sources. Additionally, the analysis shows that the use of renewable energy sources, paradoxically, is also not conducive to climate protection. This is because biomass combustion and deforestation are responsible for 10% of greenhouse gas emissions. Furthermore, less forest area means less $CO_2$ purification. At the same time, contemporary economic practice shows that legal regulations implemented by the EU and financial considerations contribute to processes unfavourable to the global climate. For example, there has been an increase in the transport of biomass from poorer to richer countries, and over long distances. Paradoxically, this increases $CO_2$ emissions. Figure 14 shows the progress towards the percentage target of renewable energy in total energy consumption. The values achieved for selected EU countries in 2012 and their target by 2020 are shown.

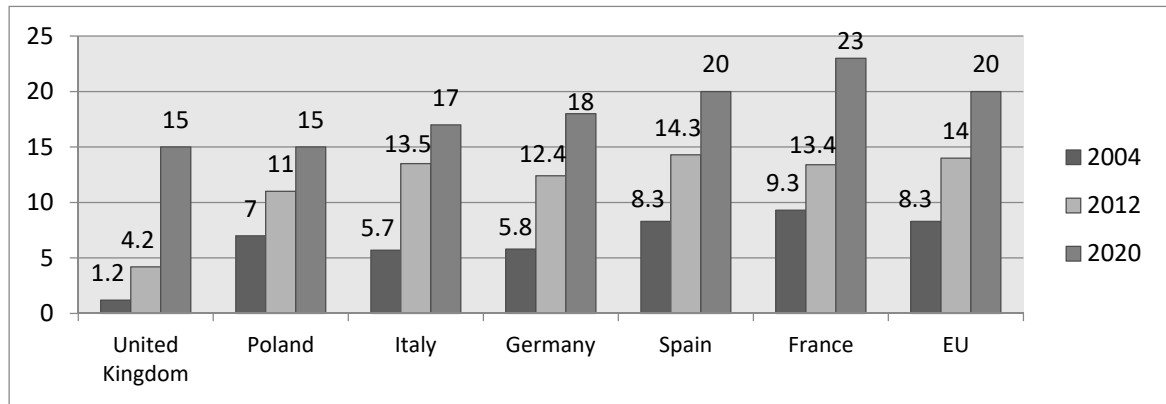

**Figure 14.** Percentage share of energy from renewable sources-implementation and percentage goals to be achieved in 2020 [30].

For the results achieved, the EU average was 14%. By 2020, EU countries still need 6% growth to meet the target (20% in 2020). However, it should be noted that the contributions achieved are not sustainable and were often obtained through costly EU subsidies. At the same time, this means that, after assessing the results, the shares will start to decline [40]. In addition, a large part of the EU countries will have problems with achieving the goals, even temporarily, and some will not achieve them (for example, the United Kingdom).

By analysing below data (Figure 15), we can see that the main sources of low-emission energy are hydropower and nuclear power plants. This shows the limitations in the use of these sources and the countries that are therefore in a worse situation having to implement the EU climate package. It should be noted that several countries in Europe already have the structure of energy sources according to the EU ideal, having 100% renewable energy or high energy values exceeding 60%, and these are: Iceland—100% hydropower and geothermal energy, Norway—95% hydropower, Switzerland—60% from hydropower and 35% from nuclear energy, Sweden—50% from hydropower and 40% from nuclear energy, Austria—62% from hydropower, Finland—25% from hydropower and 35% from nuclear energy, and France—80% from nuclear energy, as well as Slovakia, Belgium, and Slovenia. The essence of the matter, however, is shown by the fact that the leaders of low-emission energy are based on natural geographic conditions and obtained these shares before the mandatory targets for the share of renewable energy in the energy mix [41]. It follows that the natural geographic conditions (large water areas and topography) gave such opportunities to some EU countries. The rest have been invested in nuclear energy. Of the remaining countries that had a small share of RES ahead of the EU targets, only Denmark (15% change), Portugal (18%), Spain (11%), and Ireland (10%) made significant progress in the share of low-carbon sources [42]. However, all these countries have developed wind energy, which they had previously also had in their mix, so it was natural conditions that decided about the possibility of increasing this share. Germany increased the share of low-carbon energy only from 38% to 41% because, on the one hand, they made huge investments in renewable energy, and on the other hand, they gave up nuclear energy. The remaining countries without nuclear energy will have a similar share of renewable energy sources as Poland and Greece.

The below data (Figure 16) confirm the limited possibilities of using renewable energy sources, the mix of which in 70% is the combustion of wood (50%) and other biomass and waste (20%). Hydropower has a significant share (18%) and the use of wind energy is increasing (8%). The global promotion of the use of RES in the current state of technology is a contradiction because the practical management shows the lack of justification for the use of RES, where it is not profitable. In addition, the analysis shows that it is not significantly beneficial for climate protection for several reasons, namely, biomass combustion and deforestation are 10% responsible for greenhouse gas emissions, and fewer forests result in less $CO_2$ purification. With the global emission of 30 billion tons of $CO_2$, 4.5 tons of $CO_2$ per person are generated annually. A medium-sized tree absorbs around 5 kg of $CO_2$ per year. This means that it

takes as many as a thousand trees to deal with emissions generated by a single person. Additionally, forests are an essential factor of ecological balance and neutralization of pollutants, thus preventing environmental degradation. The preservation of forests is an indispensable condition for limiting soil erosion processes and regulating water relations as well as ensuring biological production, which affects the quality of human life. Additionally, the transport of biomass also releases $CO_2$ into the atmosphere. With the current conditions and regulations introduced by the EU, and for financial reasons, additional transport of biomass takes place, which increases $CO_2$ emissions.

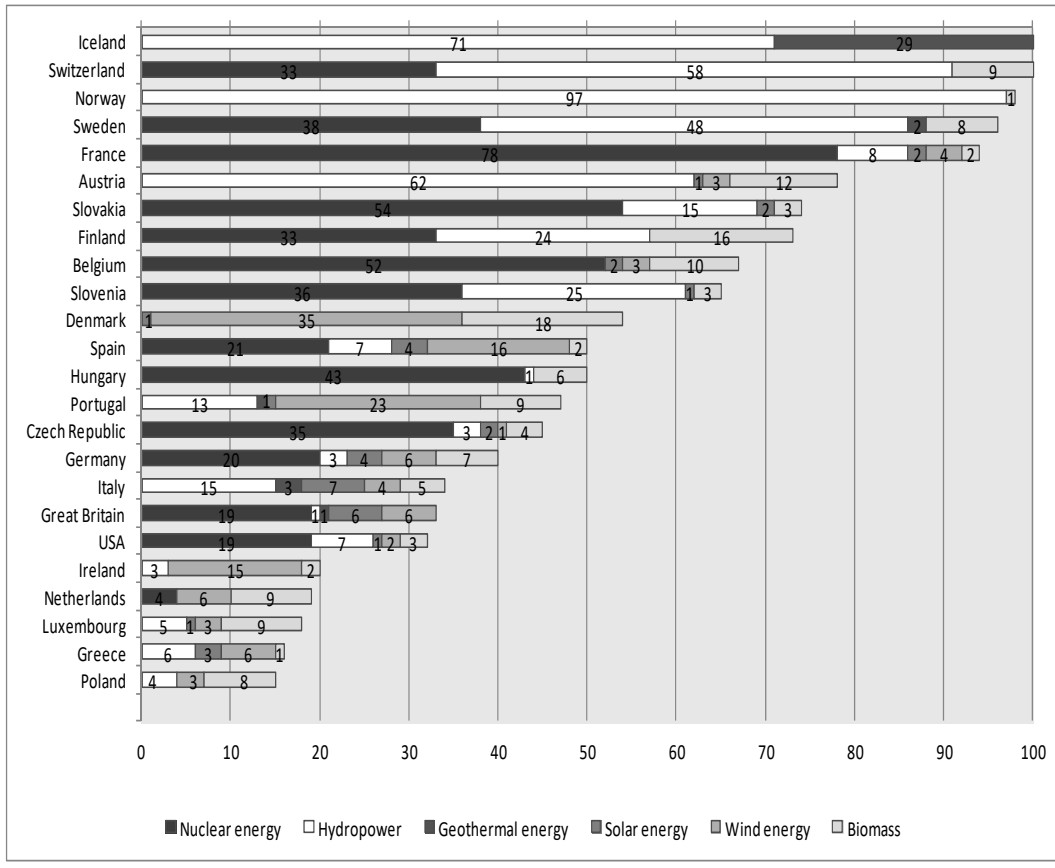

**Figure 15.** Share of low-emission sources in electricity production in 2018 (percent) [43].

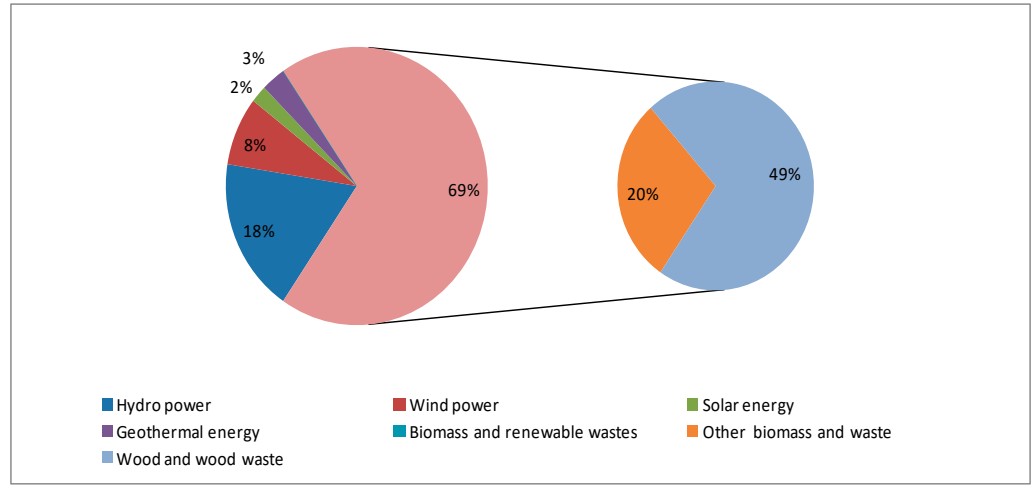

**Figure 16.** RES consumption for the EU-27 in 2018 (percent) [13].

According to the reports of the Intergovernmental Panel on Climate Change (IPCC), the main reasons for the increase in $CO_2$ concentration and the reduction of oxygen concentration in the atmosphere are energy and industrial processes as well as transport and deforestation (Figure 17).

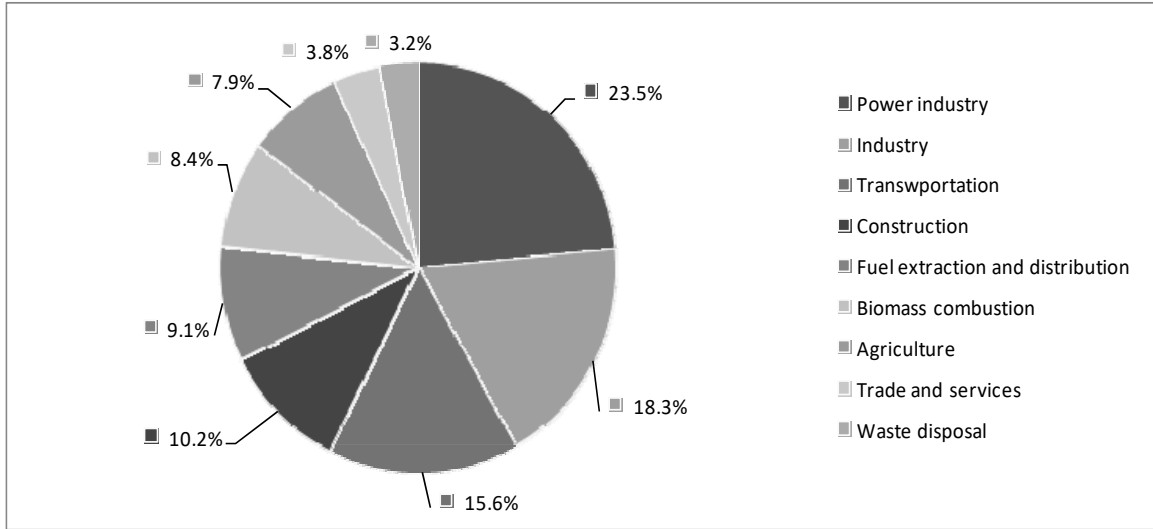

**Figure 17.** Sources of greenhouse gas emissions by sectors of the economy in 2018 [40].

## 4. Discussion

The presented analysis of the energy situation of EU Member States shows the limitations and economic harmfulness of a rapid energy reorganization for climate purposes. As part of this task, a quantitative and qualitative analysis of possible strategies and energy sources and the impact of these changes on the conditions of economic development in the European Union countries was presented. In the study, particular attention was paid to the financial and economic analysis of the adjustment processes and their effectiveness in the economic dimension. Arguments for stable and sustainable changes in the energy sector of the EU countries as well as possibilities and opportunities for using innovative technologies on the principles of sustainable development were presented. This will allow for the observance of the principles of sustainable development, which assumes appropriate and consciously formed relations between economic growth and quality of life, and care for the environment.

The EU policy of increasing the share of renewable energy and nuclear energy in the energy mix of the Member States is to affect both the dependence on imported fossil fuels and the ongoing climate change. While the intended solution seems to be a good one, there are problems with implementation and, at the same time, there are more effective alternatives. The conversion of some of the traditional energy to renewable energy is associated with enormous costs, and additionally the security of energy supply and the efficiency of its production are not satisfactory. As for nuclear energy, the EU has not taken a position and gives states a free hand to develop it, or to give it up. Therefore, it is difficult to observe a coherent approach to energy policy in the EU. For example, countries such as France, Finland, Slovakia, Bulgaria, and Romania are ardent supporters of the increasing use of uranium energy, while others such as Germany, Italy, Spain, and Austria are quite the opposite. As a result of the current climate and energy policy and problems with its implementation in practice, it can be expected that the withdrawn nuclear and coal capacities will be compensated by the construction of gas power plants, as evidenced by new long-term contracts, often concluded with Gazprom. This leads to an increasing dependence on external resources, including a politically unstable Russia. On the other hand, when it comes to climate change, it should be realized that the European Union alone cannot prevent it. In 2030, EU countries will consume about 10 percent of the total amount of energy consumed worldwide. In this regard, the EU proposes an international energy policy that actively supports European interests. However, in practice, not only will not all EU countries cope with the proposed limits on carbon

dioxide emissions to the atmosphere, it is even less likely that developing countries, needing more and more cheap energy, will be willing to cooperate with the EU on this controversial issue. It should be assumed that in the case of non-EU countries, the economic interest will be the most important, and EU countries wanting to overcome the crisis and stagnation will need to reflect on the gravity of the triad of goals on security of supply and competitiveness of the energy market, and thus the entire economy.

The goals of the EU's energy policy and strategic economic goals cannot be achieved without a thorough change in the methods of developing European infrastructure. The existing legal regulations regarding the preparation and implementation of infrastructure investments do not constitute support, and even hinder the investment process. Barriers and difficulties occurring during infrastructure investments concern legal, administrative, and social aspects. There is an urgent need to develop and adopt radical legal solutions facilitating the expansion and modernization of energy infrastructure. Experts forecast that the lack of appropriate legal regulations enabling the expansion of infrastructure may in the future result in inability to cover the demand for energy utilities, paralyze the functioning and development of the economy, trade, and transport, and thus stagnate the economy and employment. The weakness of the current way of meeting energy demand is manifested by many elements of risk: physical, economic, social, and environmental. Moreover, one should anticipate the emergence of instability, resulting from the continuation of the existing tendencies and political solutions, inadequate to the dynamically changing internal conditions and in the EU's environment. Therefore, it is difficult to be optimistic when it comes to an effective and coherent energy policy of the European Union. At present, 27 Member States do not speak with one voice on energy matters and the chances that this situation will improve in the short term by 2030 are slim. The divergence of interests results from a whole range of conditions, from economic, through social, to historical ones. In addition, due to the enormous amount of bureaucracy, decision-making and the legislative process in the EU is very slow and causes new challenges and problems. On the other hand, due to the strategic nature of the issue for the entire EU economy, the issue of a common energy policy is often discussed, becoming one of the EU's most important priorities at lightning speed, which is a positive manifestation and may raise hope for an improvement in the future. The international policy and the threat posed by aggressive actions by Russia may be a particular driving force behind changes leading to a uniform and coherent EU energy policy. The most important priorities set out in EU documents in the 2030 perspective are to ensure external supplies, where international policy and infrastructure projects play a fundamental role, with the simultaneous intensification of the use of internal EU resources, improvement of energy efficiency, creating conditions for competition on the internal energy market and counteracting carbon dioxide emissions.

## 5. Conclusions

A detailed analysis of data and effects for the task of achieving the EU average of 20% of the use of renewable sources shows that the costs of achieving this goal are high and entail a number of negative phenomena. In the first place, they cause instability in the market of energy sources. The importation of biomass from outside the EU, e.g., from Africa, Asia, and South America, is a practical consequence of EU regulations. This undermines the benefits of reducing emissions [43]. Especially that this is not a real effect and a decrease in renewable sources in the energy mix is forecast due to lower economic efficiency and currently insufficient technological solutions. This is confirmed by the introduction of subsidies to energy crops, and then the obligation to co-burn biomass with fossil fuels in power plants. Additionally, in practice, the $CO_2$ reduction balance is much less favourable, as it results from the calculations due to the emissions during processing (pellet production) and transport of biomass [29]. However, it is expected that this trend will change with technological innovations in this matter, but it is a long-term forecast. In order to achieve security of energy supply and greater competitiveness in the current geopolitical situation, they require key infrastructure investments. An example of this investment is the Nabucco pipeline project, which aims to create market choices for energy sources. At the same time, a county's own resources should be used for this purpose, but in accordance with

the principles of sustainable development, which will also be allowed by the implementation of the task of obtaining wider access to the market of energy sources, first of all, internal EU countries. Therefore, the "3 × 20" plan in the current time frame will not be so rigorously implemented and some of the assumptions have already been postponed to implementation by 2050. The demand for energy will certainly grow, so higher energy efficiency is really needed. Commissioner Oettinger said that European industry needs cheap energy to remain competitive [44]. Therefore, it is expected that the EU will retreat from some plans and legislation that increase the financial costs of using energy sources.

Regarding coal and nuclear energy, the EU has not taken specific decisions as to the numerical target, and in addition, climate policy instruments (taxes and $CO_2$ emissions trading) have a negative impact on the competitiveness of energy generation from coal on the EU market. On the other hand, the costly subsidizing of renewable energy consumes billions in subsidies that have indebted states or burden recipients. Therefore, high prices of electricity from RES, reducing the competitiveness of EU economies, may cause a return to classic energy sources, especially when their prices are reduced. This is also confirmed by the statements of EU heads of government and recent events in the European arena. The countries of Central and Eastern Europe have the highest level of implementation of commitments regarding the reduction of greenhouse gases, which is the result of, inter alia, carbon depletion, the economic crisis, and the transition to low-carbon sources, among which nuclear energy has become the main focus. Germany owes 21 percent of its implementation to having the highest investments and subsidies from the EU, and the United Kingdom has moved part of its emission production outside the country. The biggest problem in this respect is shown by Spain, Portugal, and Ireland, which, despite significant investments in wind energy, show high values increasing emissions. This is due to an increase in energy demand which is mainly met by gas. On the other hand, Greece and Poland, wanting to deal with the crisis, are currently using the cheapest energy sources, which is coal. Therefore, in the case of some of the targets, the EU realizes that these values will not be achieved, especially at the time of the current loosening of climate policy to support competitiveness and security of energy supply.

Against the background of the analysis of the state and conditions, an outline of the energy strategy can be drawn up in the form of identified threats as well as future priorities and trends that will have the greatest impact on the global and EU energy industry. The development of infrastructure within the EU is the foundation for achieving the goal of creating an integrated and competitive internal market, and in external relations for strategic supplies. The above analyses show that by 2030 there will be no major changes in the structure of the use of energy sources. On the other hand, in the following years of the 21st century, research and development of new technologies for energy generation will be of great importance for the energy situation in the world. For example, the development of clean coal technologies is associated with the need to achieve better energy efficiency of coal use, as well as to obtain better economic efficiency. Therefore, there is a need to start working out more rational and highly efficient technologies for its use. In all scenarios, fossil fuels will continue to play a dominant role in the coming decades. The decline in renewable energy prices will lead to the gradual introduction of clean energies, but the reconstruction of the global energy system is a difficult and, above all, too costly a task at the moment.

A coherent, and thus effective policy should, in the first place, focus on the resources available in the territory of the EU countries in order to ensure security of supply and competitiveness, which translates into the economic growth of the EU [45]. Competitiveness should be achieved by creating an internal energy market, but with identical market laws for all Member States, and by promoting the diversification of energy sources without excluding coal, which will remain the most important energy source in the world in the second decade of the 21st century. At the same time, as part of the implementation of this goal, and in line with the strategic triad of EU objectives, it is necessary to deviate from the too far-reaching regulations that favour financially and economically ineffective energy sources through the $CO_2$ emissions trading system, and state subsidies and guarantees for decade-long energy prices, and to implement more effective decisions in the tax system. In the current

situation, the tax system, in which the value of taxes is often greater than the price of the raw material, clearly kills the competitiveness of the EU energy market and does not support climate protection tasks in this way. This is why it is so important to develop a common and single energy market that will ensure the free exchange of energy, eliminating discrimination, and providing a level playing field for competition. In this context, there must be legal harmonization and standardization. For example, in 2015, it was planned to take the first legal steps for the Energy Union, where it may have become fundamentally important to have contacts with external suppliers of energy resources. When it comes to renewable energy, it should be said that if technological innovations significantly reduce its costs, everyone will switch to it without the need for regulation. Economic models show that in such an approach, RES will actually be the best climate solution. Therefore, regional development of energy sources should be promoted in line with the possibilities of specific regions, without imposing a single criterion, in line with the conclusion of the European Economic Congress: "Energy security and solidarity instead of an absolutely promoted doctrine" [46]. It is a pity that it was only the crisis that helped European politicians understand how much Europe can lose and how little it can gain by promoting utopian projects.

Actions should be taken to ensure sustainable development of all EU member states and to ensure energy security in a manner based on the principles of rational and efficient use of energy resources. The demand for energy resources among the EU countries is huge, which is confirmed by the fact that EU countries consume 16% of the energy produced in the world and have to import it to a large extent. Therefore, the EU climate and energy policy should take into account the specificity of energy mixes of individual countries. The European Commission should take into account the specificity of specific countries and their development strategies. Arguments such as the economic efficiency of production and the possibility of using production produced for other purposes, as well as technological development in harmony with ecology, support stable and well-thought-out changes and will lead to the path of the sustainable development of EU countries.

**Author Contributions:** Conceptualization, I.M., H.W., M.B., T.J., B.W., J.K., and R.K.; methodology, I.M., H.W., B.W., and R.K.; software, I.M., M.B., and T.J.; validation, I.M., H.W., M.B., T.J., B.W., J.K., and R.K.; formal analysis, I.M., H.W., B.W., and J.K.; investigation, I.M., H.W., M.B., T.J., and R.K.; resources, I.M., H.W., M.B., B.W., J.K., and R.K.; data curation, I.M., H.W., M.B., T.J., B.W., J.K., and R.K.; writing—original draft preparation, I.M., H.W., B.W., J.K., and R.K.; writing—review and editing, M.B., T.J., and I.M.; visualization, I.M., H.W., M.B., and J.K.; supervision, I.M., H.W., J.K., and R.K.; project administration, I.M., H.W., and J.K.; and funding acquisition, I.M., H.W., B.W., and J.K. All authors have read and agreed to the published version of the manuscript.

**Funding:** This research was funded by: Natolin European Center grant for "summer research" at the Library of the European University Institute in Florence. The project was financed within the framework of the program of the Minister of Science and Higher Education in Poland under the name "Regional Excellence Initiative" in the years 2019–2022, project number 001/RID/2018/19, the amount of financing PLN 10,684,000.00.

**Acknowledgments:** Many thanks to Teresa Lubińska, Beata Filipiak, and Leon Dorozik for scientific support and Justyna Miciuła for spiritual support.

**Conflicts of Interest:** The authors declare no conflict of interest.

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
