# Peer review of "Management of the Energy Mix and Emissivity of Individual Economies in the European Union as a Challenge of the Modern World Climate†"

_energies, doi:10.3390/en13195191_

Round 1

Reviewer 1 Report

The authors attempted to present the most important elements to be implemented in the EU energy policy in the 2030 perspective in the context of sustainable development of the Member States. The authors have been quite successful in achieving their goal. However, I have some comment, in any case minor.

- Although it is true that the manuscript is purely descriptive / discursive and the authors have not used any statistical inference method, they could point out some limitations: access to data, different definitions, time period, etc.
Therefore, authors should add a paragraph of limitations and, perhaps, another of strengths.

Author Response

Thank you very much for making a substantive and helpful review. The introduction was corrected. The discussion point was corrected and the conclusions rebuilt. The article has been supplemented with a broader overview.
Thanks again for pointing out possible corrections.
The corrected article is attached.

Reviewer 2 Report

General comment:

The manuscript deals with the presentation, according to the authors, of the most important elements to be implemented in the EU energy policy in the 2030 to achieve sustainable development of the Member States. In particular, the manuscript is focused on the energetic strategy to be implemented.

The manuscript is suitable to be published in this journal, however some points should be addressed before publication. In particular, the manuscript sounds like a technical report and not like a review article.

Some minor language mistakes are present that should anyway be corrected.

  1. Literature Review

Please, include a short overview of some technical approaches that can be used for the reduction of the CO2 emission. For example, CO2 capture by microalgae is a potential methodology that not only allows to avoid CO2 emission, but also to produce valuable by-products. Please, consider the following manuscript:

  • Bench-scale cultivation of microalgae scenedesmus almeriensis for CO2 capture and lutein production (2019) Energies, 12 (14), art. no. 2806. DOI: 10.3390/en12142806
  • Techno-Economic Study of CO2 Capture of a Thermoelectric Plant Using Microalgae (Chlorella vulgaris) for Production of Feedstock for Bioenergy (2020) Energies, 13 (2), art. no. 413. DOI: 10.3390/en13020413
  • Carbon dioxide capture with microalgae species in continuous gas-supplied closed cultivation systems (2020) Biochemical Engineering Journal, 163, art. no. 107741. DOI: 10.1016/j.bej.2020.107741
  • Carbon dioxide capture and use in photobioreactors: The role of the carbon dioxide loads in the carbon footprint (2020) Bioresource Technology, 314, art. no. 123745. DOI: 10.1016/j.biortech.2020.123745
  1. Results

Please, quantify CO2 emission reduction according to the proposed strategy.

Please, highlight the role of renewable energy. In particular, quantify the energy produced from renewable sources.

Figure 6: please, parameter and unit on ordinate axes.

  1. Discussion

The manuscript sounds like a technical report and not like a review article.

The relationship between the section “Results” and the section “Discussion” is not very clear from my point of view by considering the results that have been shown.

  1. Conclusions

Conclusions seem not be supported by the results shown.

It is not clear how this statement “In practice, the CO2 balance is significantly less advantageous than what follows from theoretical calculations due to emissions during processing (production of pellets) and transportation of  biomass” is supported by the presented results, while you included a reference.

Author Response

Thank you very much for making a substantive and helpful review. The introduction was corrected. The discussion point has been corrected and the conclusions have been rebuilt. Figure 5 has been corrected because the country names have been folded up when editing. Other non-EU countries in the analysis data were selected for comparison purposes only. Linguistic errors were corrected and the article was supplemented with a broader review. The proposed literature was taken into account and added, which extends the analysis performed.

Thanks again for pointing out possible corrections.

The corrected article is attached.

Reviewer 3 Report

Recommendation: Interesting paper. My advice would be to ask for a major revision of the manuscript.

Comments:

The manuscript entitled “Energy mix and emissivity of individual economies in the European Union as a climate challenge for the modern world” presents an interesting work on introducing important elements of EU energy policy and the perspective on climate challenge for EU member states. Overall, it has some merits.

The literature review is good. I like the comparison between each EU member. This would give us how the EU energy policy would change the climate for each member. However, the introduction is weak. The current introduction has not been geared toward the aim of this article. What is the basis of this policy, and why use Figure1 as an example? Can it represent all other members? In Figure 5, there are more than 10 members listed and they are all different. The authors need to pay more attention to focus on how the EU polity may impact the overall EU and individual EU member.

Another comment is: from Figure 5 – 10, different EU members were selected and non-EU members were randomly selected as well. Are there any particular reasons for selecting other countries instead of EU members? This should be very clear in the result section as well as the discussion.

Author Response

Thank you very much for making this helpful review. The introduction, discussion and conclusions section have been corrected.
Figure 5 has been corrected because it collapsed the names of the countries when editing and reducing it. Other non-EU countries in the analysis data were selected for comparison purposes only. Thanks again for pointing out possible corrections. The corrected article is attached.

Round 2

Reviewer 2 Report

The authors modified the manuscript according to my suggestione, the paper can be published.

Reviewer 3 Report

The manuscript may be accepted for publication.